# The Mediating Effect of Nature Restorativeness, Stress Level, and Nature Connectedness in the Association between Nature Exposure and Quality of Life

**DOI:** 10.3390/ijerph19042098

**Published:** 2022-02-13

**Authors:** Migle Baceviciene, Rasa Jankauskiene

**Affiliations:** 1Department of Physical and Social Education, Lithuanian Sports University, Kaunas LT-44221, Lithuania; 2Institute of Sport Science and Innovations, Lithuanian Sports University, Kaunas LT-44221, Lithuania; rasa.jankauskiene@lsu.lt

**Keywords:** natural environment, restorativeness, connectedness to nature, distress, quality of life

## Abstract

This study aimed to test the mediating effects of nature restorativeness, stress, and nature connectedness in the association between nature exposure and quality of life (QoL). Urban and rural Lithuanian inhabitants (n = 924; 73.6% were women), mean age of 40.0 ± 12.4 years (age range of 18–79) participated in the study. In total, 31% of the respondents lived in rural areas. Study participants completed an online survey form with measures on sociodemographic factors, nature proximity, nature exposure, nature connectedness, and nature restorativeness, stress, and QoL assessed by the abbreviated version of the World Health Organization’s Quality of Life Questionnaire’s (WHOQOL-BREF). Path analysis was conducted to test the mediating effects of nature restorativeness, stress, and nature connectedness in the model of nature exposure and QoL. Nature exposure was directly associated with a greater QoL (β = 0.14; *B* = 2.60; SE = 0.57; *p* < 0.001) and mediated the association between nature proximity and QoL. Nature restorativeness and lower stress levels were mediators between nature exposure and QoL. Nature connectedness was a mediator between nature exposure and QoL. A path model was invariant across genders and the urban and rural place of residence groups: patterns of loadings of the pathways were found to be similar. Nature restorativeness (β = 0.10–0.12; *p* < 0.01) had a positive effect on the psychological, physical, social, and environmental domains of QoL. Connectedness to nature positively predicted psychological (β = 0.079; *p* < 0.05) and environmental (β = 0.082; *p* < 0.05) domains of QoL. Enhancing nature exposure and nature connectedness might help strengthen QoL in urban and rural inhabitants.

## 1. Introduction

### 1.1. Nature Exposure and Quality of Life

Modern urban life is considered stressful because it is related to overcrowding, traffic, excess information, noise, and lack of natural surroundings [1,2]. Stress can exert various actions on the body, ranging from alterations in homeostasis to life-threatening effects and death [3]. Stress is defined as the process by which an individual responds psychologically, physiologically, and often with behaviors, to a situation that challenges or threatens well-being [4]. Studies show that stress is negatively associated with quality of life (QoL) [5,6]. 

A greater amount of nature and natural environments is related to enhanced public health in urban areas of economically developed countries [7]. The term “natural environment” refers to the continuum of environments from wild nature to designed green spaces [8]. Studies report that nature exposure is associated with the nature proximity. Specifically, results of previous studies showed that availability of nature and green spaces in a living environment is associated with higher nature contact, especially in high urbanization cities [9,10]. 

Mounting evidence suggests that nature exposure has numerous effects on physiological and psychological health, such as reduced stress, decreased blood pressure, enhanced immune system resources, increased physical activity, enhanced positive body image, lower depression and anxiety, better quality of sleep, happiness, vitalizing effects, and restored cognitive function [11,12,13,14,15,16]. Exposure to natural environments is associated with affective benefits such as the reduction of stress and negative effects and an increase in positive effects and subjective well-being [4]. 

Although evidence constantly shows positive associations between nature exposure and subjective well-being, less is known about the relationships between nature exposure and quality of life. This might be explained by the fact that, in the research literature, definitions of well-being, happiness, and QoL are highly overlapping [17]. Well-being was described as a state of positive feelings and meeting full potential in the world [18]. Much cultural variation exists in the concepts of happiness, with many linguistic traditions centering on fortune or luck or an individuals’ self-perceived success, which is an aspect of life satisfaction [17,19]. The World Health Organization (WHO) defines QoL as an individual’s perception of their position in life in the context of the culture and value systems in which they live and in relation to their goals, expectations, standards, and concerns [20]. QoL involves psychological, physical, social, and environmental domains. Recent evidence suggests that all these domains together explain 73% of the variance in happiness and 66% of subjective well-being [17]. However, the environmental domain explained only 14% of the variance in happiness and was not a significant predictor of it, while the psychological domain was the strongest predictor of happiness and subjective well-being. Therefore, it was suggested that only the psychological domain of QoL might be interchangeably used with happiness and subjective well-being [17]. Nevertheless, the physical, social, and environmental domains might also be important when analyzing associations between nature exposure and QoL. Notably, there is lack of empirical data demonstrating the link between nature exposure and various domains of QoL. Furthermore, the mechanisms that help explain the associations between nature exposure and QoL are not well explored. Understanding how nature exposure is associated with specific outcomes of public health importance, such as QoL, is one of the proposed research priorities [4,21].

### 1.2. Nature Restorativeness, Connectedness to Nature, Stress, and Well-Being

Evidence exists that natural environments are perceived to be more restorative than urban environments [22]. Environmental psychology theories, such as Psychophysiological Stress Recovery Theory (SRT) [23] and Attention Restoration Theory (ART) [24], have been used to explain the associations between exposure to nature and its outcomes on human well-being. Psychophysiological theory SRT suggests that contact with nature might rapidly increase positive emotions in a person experiencing acute stress. Positive effect, in turn, blocks negative thoughts and reduces physiological activation and stress. According to the SRT, exposure to nature has an immediate stress reducing effect and helps human beings prepare themselves for various future life-tasks. 

In contrast, ART states that the modern world places a demand on a human’s cognitive and emotional systems for which they are not necessarily well adapted [24]. On the other hand, environments with high restorative potential (i.e., nature environments) provide humans with opportunities for psychological restoration and resting inhibitory mechanisms on which attention depends and facilitate better recovery from mental fatigue compared to urban environments [24,25]. The SRT states that recovery from stress is necessary for mental restoration to occur. In contrast, the ART states that mental resource depletion might explain the increase in physiological stress and resource replenishment can better explain attention restoration. In other words, ART posits that natural environments have ability to restore the depleted mental resources, such as cognition, attention, and emotions, that often results from the negative factors of urban environments [24,25]. Thus, ART and SRT theories differ in explanation of what drives and individual to seek a restorative place: ART states that it is recovery from mental fatigue, whereas SRT proposes recovery from stress. Nevertheless, recent findings of a neuroscience-based study suggested that mental restoration and stress recovery co-occur and that they are bidirectional manifestations of activity in the vagus nerve, which links the peripheral nervous system to the central nervous system [26]. 

To the best of our knowledge, nature restorativeness and stress reduction has never been tested as the mediators between nature exposure and QoL; therefore, one of the objectives of the present study is to fill that gap. Thus, one of the objectives of the present study was to test the mediating roles of nature restorativeness and stress reduction in the associations between nature exposure and greater QoL.

Some researchers claim that focusing only on restorative effects of nature is too simplistic because the benefits of the relationship between humans and nature go well beyond the acute restorative effects induced by nature exposure [27]. Based on the Biophilia hypothesis, it was suggested that, driven by biological evolution, human beings have an innate affinity with being around other living creatures [28]. Connectedness to nature is human’s conscious feelings of being part of the natural world, as opposed to feeling separate from it. In other words, connectedness to nature is the extent to which people identify themselves with non-human living things [29,30]. It is composed of cognitive, affective, and behavioral components [31]. 

Evidence suggests that connectedness to nature is positively related to life satisfaction, higher self-esteem, eudaimonic well-being, mindfulness, and meaning in life [30,32,33,34,35,36,37,38,39,40]. Connectedness to nature might help shift humans’ attention from consumerism-based concerns to more holistic experiences and to promote an attitude of ecocentric connections, where humanity is seen as part of a global ecosystem [29,41]. For example, evidence suggests that connectedness to nature is negatively associated with problematic smartphone use [42] and positively associated with environmental identity and ecological behavior [43]. Recent meta-analysis concluded that individuals who are more connected to nature tend to have greater eudaimonic well-being and have higher levels of self-reported personal grow [38]. Further, based on the self-determination theory (SDT) [44], nature exposure might fulfill one of the basic human needs, specifically—relatedness to others—through increased feelings of non-human types of relatedness [45,46]. Fulfilling basic psychological needs, such as autonomy, relatedness, and competence, is associated with the greater psychological well-being [46]. 

Humans have different levels of connectedness to nature [21,37]. Findings of recent studies suggested that nature connectedness mediate the associations between nature exposure and well-being [47,48]. Evidence exists that nature connectedness impacts how individuals respond to natural environments. A recent study found that connectedness to nature is a moderator between nature exposure and positive effects. Specifically, individuals high in nature connectedness reported higher levels of positive affects to the natural versus built environments [49]. Nature connectedness is higher in people who have previous experience of nature [30,50]. However, experimental studies showed that trait levels of nature connectedness did not moderate nature’s beneficial impact on well-being suggesting that nature involvement is beneficial among a variety of individuals [51,52]. However, the mediating role of connectedness to nature in the associations between nature exposure and QoL is rarely explored, therefore the present aimed to do so. Based on previously presented evidence, it is reasonable to expect that connectedness to nature will mediate the associations between nature exposure and QoL. 

Finally, studies have revealed an association between connectedness to nature and increased feelings of psychological restoration [49,53]. The associations between connectedness to nature and nature restoration might be bidirectional. Specifically, some studies showed that the restorative effects of nature are the result of an individual’s level of connectedness [47]. However, findings of other studies suggested that an individual’s connectedness to nature might be influenced by the extent to which they found their experiences of nature restorative [49,54]. Thus, it is important to further explore this issue.

### 1.3. The Present Study

This study aimed to test the mediating effects of nature restorativeness, stress, and nature connectedness in the association between nature exposure and quality of life (QoL). The second aim of the present study was to test mediating role of nature exposure in the associations between nature proximity and QoL. For this purpose, we developed a theoretical model (Figure 1). Based on the previous findings, we expected that nature exposure would mediate associations between nature proximity and enhanced QoL. Furthermore, we expected that nature restorativeness and lower stress level would mediate the associations between nature exposure and enhanced QoL on the one hand and nature connectedness on the other. Next, based on the previous findings we expected that lower stress level would mediate the associations between nature exposure and QoL. A recent systematic review reported that the place of residence might be an important moderator in the associations between nature exposure and physical health [55]. Therefore, for exploratory purposes, we also assessed the extent to which the final model was invariant across urban and rural residents. Finally, we aimed to test predictive power of nature exposure, nature restorativeness, nature connectedness, and stress level on various areas of QoL. We hypothesized that the study variables would positively predict all four domains of QoL (psychological, physical, social, and environmental). 

## 2. Materials and Methods

### 2.1. Procedure

The present study was a part of a more extensive international study “Body Image and Nature Survey (BINS)” [56]. The study was approved by the Social Research Ethics Board of Lithuanian Sports University (protocol number SMTEK-60, 24 November 2020). The study was implemented through the Google Forms survey platform. Study participants were recruited by a non-probabilistic volunteer sampling method, and their participation was voluntary without providing remuneration. Higher response rate in women than in men resulted in a sample disproportion across gender groups. A link to the survey was shown as a sponsored advertisement on Facebook covering the main country municipalities and inviting to participate women and men from 18 years. In addition, social networking of the local public health bureaus was used to spread information about the study and link survey.

Inclusion criteria were set for age (18 and over) and language spoken (Lithuanian). Prior to completing the survey, participants were introduced to the study aims and the average duration to complete the form (about 25–30 min.). In addition, information about survey anonymity was provided. Study participants could provide their digital consent to participate or could decline to participate. After providing digital consent, responders were provided the study measures. Those who declined to participate were acknowledged, and the survey was terminated. Moreover, the online survey could be stopped at any point by closing the browser without recording the answers. Nine persons refused to participate. The final study sample consisted of 924 adult Lithuanian men and women containing no missing data, as all study questions were mandatory.

### 2.2. Study Participants

Using the continuously varying sample size approach to Monte Carlo power analysis, approximately 150 individuals were required to ensure the statistical power is at least 80% for detecting the hypothesized indirect effect [57]. Statistical power is the probability of rejecting null hypothesis (H_0_) given the alternative one (H_1_) is true, if one can draw a large number (e.g., 5000) of random samples (replications) from the population defined by H_1_ and fit the hypothesized model on the samples. A power can be estimated as r/R, the number of samples that reject H_0_ (r) divided by the total number of samples (R) [57]. In our study, for the multiple serial model, a power of 0.80 can be achieved with a sample size n of 750. The calculated power for the sample size n of 900 was 0.88 (95% CI 0.84–0.91).

The present study was comprised of 924 Lithuanian inhabitants, 680 (73.6%) women and 244 (26.4%) men with an age range of 18–69 years (M = 40.0; SD = 12.4). Regarding their area of residence, 69.9% of the participants resided in an urban area (64.8% of men and 71.8% of women), while 30.1% lived in a rural one (35.2% of men and 28.2% of women). Most participants were married (57.8%), 18.4% were single, and 17.2% were in a long-term relationship (6.6% did not specify their marital status). In terms of educational qualifications, 11.3% of the participants had completed secondary education or less, 7.7% were in full-time education, 41.7% had completed an undergraduate degree, and 34.8% had a postgraduate degree (4.4% of the study respondents did not specify their educational attainment). The ethnic majority comprised 91.1% of the sample, while 2.9% assigned themselves to ethnic minority group. The remaining 6.0% did not indicate their ethnic group. All the respondents could speak and understand Lithuanian language.

### 2.3. Measures

First, respondents were asked to provide answers for sociodemographic data (age, gender, education, place of residence, marital status, ethnicity, height, and weight). The place of residence was classified into two groups: urban (capital, cities, towns) and rural (rural areas and suburbs).

The Nature Exposure Scale (NES) [58] consists of four questions asking participants (1) to rate the level of exposure to natural environments with possible answers on Likert type scale ranging from 1 (very little of my everyday natural environment is natural) to 5 (most of my everyday environment is natural); (2) to indicate if the respondent notices the natural environments in his/her everyday life (possible answers range from 1 (not much) up to 5 (a great deal); (3) to rate the frequency of exposure to nature-rich environments outside of his/her everyday environment (possible answers range from 1 (once a year or less to 5 (once a month or more often); (4) to evaluate how much he/she takes notice of the nature in natural environments (possible answers range from 1 (not much) to 5 (a great deal). Answers were averaged and higher scores indicated greater nature exposure. The Lithuanian version of the scale showed acceptable psychometric properties: Cronbach’s α was 0.70 and unidimensional factor structure with the adequate model fit indices was confirmed (CFI = 0.96; RMSEA = 0.09) [59]. For this study, Cronbach’s α was 0.69.

The Restoration Outcome Scale (ROS) [60] measures restorative outcomes after most recent contact with natural environments. The scale measures the degree of restorative outcomes after most recent visit to a natural environment in terms of calmness, relaxation, attention, restoration, clarity of thought, subjective vitality, and self-confidence (sample item: “During my most recent visit to a natural environment I felt restored and relaxed”). The scale consists of nine statements with a Likert type response scale ranging from 1 (not at all) to 7 (completely). An overall score was calculated by averaging the response options. Higher scores indicate greater restoration after the most recent visit to a natural environment. The Lithuanian translation of the scale showed acceptable psychometric properties: Cronbach’s α was 0.98 and unidimensional factor structure with the adequate model fit indices was confirmed (CFI = 0.99; RMSEA = 0.09) [59]. In the present study, Cronbach’s α for the ROS items was 0.98.

The Connectedness to Nature Scale (CNS) [30] is a 14-item instrument that measures an individual’s affective and experiential connection to nature (sample item “I often feel a sense of oneness with the natural world around me”). Answers are rated on a 5-point scale from 1 (strongly disagree) up to 5 (strongly agree). An overall score is calculated as the mean of the response options. A greater score reflects greater connectedness to nature. The Lithuanian version of the scale demonstrated good psychometric properties: a unidimensional factor structure with the adequate fit indices was confirmed (CFI = 0.97; RMSEA = 0.07) and Cronbach’s α was 0.90 [59]. In the present study, Cronbach’s α for the CNS was 0.90.

The Reeder Stress Inventory (RSI) was used to assess trait psychological stress [61]. In the present study, we used an adapted seven-item RSI that was validated for the Lithuanian language [62]. The participants were asked to indicate the answer for each of the statements describing their general stress-related feelings (sample item: “In general, I am nervous”). Responses were based on a 4-point Likert type scale from 1 (yes, I agree) to 4 (no, I disagree). The scores for all items were summed, and the higher rating on the scale indicates lower perceived stress. In the present sample, the Lithuanian version of the scale demonstrated good psychometric properties: a unidimensional factor structure with the adequate fit indices was confirmed (CFI = 0.99; RMSEA = 0.08), the internal consistency (Cronbach’s α) was 0.85.

The Quality of life (QoL) questionnaire was used to assess QoL [63,64]. This questionnaire is an abbreviated version of the World Health Organization QoL 100 (WHOQOL-100) questionnaire consisting of 26 items. Then, 24 items of the instrument comprise four separate domains (physical, psychological, social, and environmental) with the responses rated from 1 (very dissatisfied) to 5 (very satisfied). The physical domain consists of six statements (sample item: “To what extent do you feel that physical pain prevents you from doing what you need to do?”), the psychological domain consists of 7 (sample item: “How much do you enjoy life?”), the social domain consists of 3 (sample item: “How satisfied are you with your personal relationships?”), and the environmental domain consists of 8 items (sample item “How satisfied are you with the conditions of your living place?”). The scores are transformed into a scale ranging between 0 and 100, where 0 is very poor and 100 is very good. Two questions assessing the overall understanding of health and overall QoL were evaluated separately. The Lithuanian version of the instrument showed acceptable psychometric properties [65]. In the present study, the internal consistencies of the physical, psychological, social, and environmental domains represented by Cronbach’s α were 0.79, 0.83, 0.80, and 0.80, respectively. For the whole scale, Cronbach’s α was 0.92.

Nature proximity was assessed by asking participants to report the distance from their living place to the nearest natural environment (park, forest, lakes, sea, or other natural environments). Possible answers were classified into ranges: 10 km and more, 5–10 km, 2–4 km, 1–2 km, 0–1 km, and 0 km (nearby).

### 2.4. Statistical Analyses

Preliminary analyses were performed using the IBM SPSS Statistics v. 27.0 software (IBM Corp., Armonk, NY, USA) and AMOS (Analysis of Momentary Structure) v. 26 for the path analysis (IBM Corp., Armonk, NY, USA). Internal consistency of the scales was tested by Cronbach’s α coefficients. The values of Cronbach α ≥ 0.7 was considered as adequate, ≥0.8 as good, and ≥0.9 as excellent. Before the analysis, all study measures were tested for normality and outliers by evaluating skewness, kurtosis, and the Q-Q plots. All normality indices were confirmed with no outliers detected allowing further parametrical tests (e.g., *t*-test) application.

Differences in the study measures between the two groups (urban and rural citizens) were tested through *t*-tests, and the effect size was calculated through Cohen’s d with the Hedges’ correction for unequal sample sizes. Effect sizes above 0.2 were considered small, equal or above 0.5 were moderate, and equal or above 0.8 were strong [66]. To explore the associations between nature proximity, nature exposure, nature connectedness, nature restorativeness, stress, and quality of life, Pearson correlation analyses were run separately in urban and rural groups; magnitudes between 0.1 and 0.3 were considered small, above 0.3 and below 0.5 were moderate, and equal or above 0.5 were strong with a significance level of <0.05 [67].

Finally, to test the mediating effects of nature connectedness and nature restorativeness in the association between nature exposure, stress, and QoL, a path analysis was conducted to estimate the presumed relations within the hypothesized model. The bootstrap approach was used to conduct mediation analyses with 5000 bootstrap samples drawn from the dataset to calculate indirect and direct effects and bias corrected 95% CIs [68]. The 95% CIs for the coefficients calculated by the bootstrapping methods were considered statistically significant if the confidence intervals did not include zero. Model fit was assessed using indices recommended by Hu and Bentler [69]: the normed model chi-square (χ2/df; values < 3.0 considered indicative of a good fit), the standardized root mean square residual (SRMR; values < 0.09 indicate a reasonable fit), the comparative fit index (CFI; values close to or >0.95 indicate an adequate fit), and the root mean square error of approximation (RMSEA) and its 90% CI (values close to 0.06 indicative of good fit and values up to 0.08 indicative of adequate fit).

In addition, with the intent to test model invariance across urban/rural and men/women groups, a multi-group analysis was performed. The assumption about configural invariance (pattern of loadings of the pathways on indicators) of the final model across groups (urban vs. rural, women vs. men) was tested.

## 3. Results

No significant differences in study measures were found comparing study variables in men and women groups. The results indicated that the urban and rural groups did not present significant differences concerning nature connectedness, nature restorativeness, stress, and QoL, with the exception of physical domain of QoL. However, the results showed that participants from the urban group had significantly lower means of nature proximity and nature exposure when compared with the rural group (Table 1). Mean scores of the physical domain of QoL were higher in the urban group compared to the rural group. Considering Cohen’s guidelines to discuss the effect size of these differences, all variables presented small effect sizes with significant differences. 

The results in Table 2 demonstrated that nature proximity was not significantly associated with nature restorativeness and nature connectedness in the urban group and positive associations between these variables were observed in the rural group. In addition to the fact that all other study variables were positively inter-correlated at a weak or moderate level, it is important to note that correlations were stronger in the rural citizens group (r = 0.15–0.63) compared to the urban group (r = 0.06–0.55).

Multiple linear regression analyses examining the effects of nature-associated scales and stress on different QoL domains were performed (Table 3). Before the analysis, data were tested for possible multicollinearity effect. Variance inflation factors (VIFs) ranged in the interval between 1.07 to 1.64 allowing to keep all the variables in the models.

Nature restorativeness and lower stress level predicted better ratings of the QoL in all domains. Importantly, nature proximity had an independent significant effect on environmental QoL domain. However, nature exposure had no impact on satisfaction with social relationships domain, whereas the effect of nature connectedness was significantly associated with the improved QoL only in psychological and environmental domains.

Figure 2 represents the final path model. The hypothesized model demonstrated adequate model fit indices, χ2 = 26.554; *p* < 0.001; *d*f = 6; CFI = 0.981; RMSEA = 0.061 (90% CI 0.038–0.085). All path coefficients were significant (*p* < 0.01) with a positive valence.

There were direct effects from nature proximity to nature exposure (estimate = 0.15; SE = 0.02; *p* < 0.001), from nature exposure to nature restorativeness (estimate = 0.24; SE = 0.06; *p* < 0.001), and from nature exposure to nature connectedness (estimate = 0.43; SE = 0.03; *p* < 0.001). Moreover, nature exposure had a positive impact on the lower stress level (estimate = 1.23; SE = 0.20; *p* < 0.001). Next, nature exposure (estimate = 2.60; SE = 0.57; *p* < 0.001) and nature connectedness (estimate = 2.63; SE = 0.57; *p* < 0.001) positively affected QoL whereas nature restorativeness had a positive effect on reduced stress level (estimate = 0.27; SE = 0.10, *p* < 0.01). Finally, lower stress level was positively associated with the QoL (estimate = 1.57; SE = 0.09; *p* < 0.001).

Together with the direct estimates, Table 4 provides mediated effects. There were significant serial mediations from nature proximity, nature exposure via lower stress level, and nature connectedness to QoL. In addition, there were significant serial mediations from nature proximity via nature exposure to nature connectedness, nature restorativeness, and lower stress level. Moreover, there was an indirect effect from nature exposure to nature restorativeness via nature connectedness. Nature restorativeness and lower stress level mediated the association between nature exposure and QoL.

Next, we assessed configural invariance of the final model across gender and place of residence groups (urban and rural). The results showed that the model fitted the data across gender groups, χ2 = 34.553; *p* < 0.001; *d*f = 12; CFI = 0.980; SRMR = 0.031; RMSEA = 0.045 (90% CI = 0.028–0.063), as well as across urban and rural resident groups, χ^2^ = 32.000; *p* < 0.001; *d*f = 12; CFI = 0.982; SRMR = 0.029; RMSEA = 0.043 (90% CI = 0.025–0.061). 

## 4. Discussion

The aim of the study was to broaden the current understanding of the relationships between nature exposure and QoL. In the present study, we developed a theoretical model of nature exposure and QoL with the aim of testing the mediating effects of nature restorativeness, stress, and nature connectedness in the association between nature exposure and enhanced QoL. The second aim of the present study was to test the mediating effect of nature exposure in the associations between nature exposure and QoL. We expected that nature exposure would mediate the associations between nature proximity and the greater QoL and nature restorativeness, and reduced stress will mediate the associations between nature exposure and QoL on the one hand and connectedness to nature on the other. 

The developed model demonstrated adequate fit to the data. The results of the path model suggested that nature exposure mediated associations between nature proximity and enhanced QoL. These results are in line with findings of the previous studies, suggesting that availability of nature and green spaces in a living environment is associated with the higher nature contact, especially in areas of high-urbanization [9,10]. However, the important novel finding of the present study is that nature proximity is associated with enhanced QoL via nature exposure. Previous studies reported that living in a greener neighborhood was unrelated to well-being, however the present study adds to the knowledge that nature proximity might be associated with enhanced QoL through nature exposure [36]. These findings are consistent with previous evidence from Western countries demonstrating associations between nature exposure and well-being [4,70]. However, as discussed previously [17], QoL is a broader construct than well-being; thus, the present study adds to the knowledge that nature exposure is associated not only with psychological but also other domains of QoL, such as physical or environmental. 

Further, the results of the path model showed that nature restorativeness and lower stress mediated the associations between nature exposure and QoL. This finding is in line with the main tenets of the ART, suggesting that exposure to nature reduces stress through restoration of mental resources [24]. The mediating role of nature restorativeness was observed between the perceived biodiversity of nature and emotional well-being [71] and between nature-based recreation and emotional well-being [72]. However, since the present study is one of the first, further investigation is required to provide a deeper understanding of the mediating role of nature restorativeness and stress in the associations between nature exposure and QoL. 

In line with our next assumption, we observed that lower stress level was a mediator in the association between nature exposure and QoL. These results are in accordance with assumptions of SRT that suggest direct associations between nature exposure and stress reduction [23]. Previous studies also showed that stress is negatively associated with quality of life (QoL) [5,6]. 

Further, the present study contributes empirical evidence that connectedness to nature is a mediator between nature exposure and QoL. Our findings extend the prior theory and research, and are in accordance with the previous studies that demonstrated state connectedness to nature was a mediator in the association between nature exposure and well-being [47,48,73]. The novel finding of the present study is that nature exposure relates to higher general QoL, both directly and through connectedness to nature. The results of the present study underscore the importance of the connectedness to nature to enhanced QoL. 

Finally, we observed the mediating role of nature connectedness in the associations between nature exposure and nature restorativeness. Previous studies have revealed an association between connectedness to nature and increased feelings of psychological restoration [49,53]. Thus, the present study adds empirical knowledge that nature exposure might increase feelings of mental restorativeness via nature connectedness. However, since this study is cross-sectional, the associations might be bidirectional. Some studies showed that the restorative effects of nature are the result of an individual’s level of connectedness [47] and an individual’s connectedness to nature might be influenced by the extent to which they found their experiences of nature restorative [49,54]. Therefore, future studies of experimental and longitudinal designs must be implemented to deepen understanding on this issue. 

In the present study, we found that the newly developed model of nature exposure and QoL was invariant across urban and rural residents. However, despite the existing evidence that place of residence might be an important moderator in the associations between nature exposure and physical health [55], our study found no evidence for the differences between nature restorativeness and nature exposure in urban and rural inhabitants. However, as expected, rural residents reported significantly greater nature proximity and exposure. Notably, positive associations between nature proximity, nature restorativeness, and nature connectedness were only observed in the rural group, and all study variables were more strongly correlated in the rural inhabitants group. It is important to note that despite the fact that rural inhabitants reported more natural surroundings in their living area compared to urban residents, connectedness to nature and nature restorativeness were similar in the present study. It is a novel finding, and other studies should provide more data on this issue. However, living place should be considered an important variable in future studies assessing nature exposure and well-being. 

Finally, multiple linear regressions showed that nature exposure was the most important predictor of physical, psychological, and environmental domains of QoL. However, neither nature exposure nor nature proximity predicted social domain of QoL. This might be explained by findings that nature might not only enable pleasant social contacts but also might be seen as the place for escaping social contacts, and especially social pressures [70]. However, this result might also show that people do not relate visiting nature with the general quality of social relationships. Nevertheless, our findings are novel, and the results might be random; therefore, future studies should provide more data on this issue. 

Notably, regression analyses showed that nature restorativeness had an effect for all domains of QoL. This finding adds to the knowledge that restoration in nature is important not only for psychological well-being but also for physical, social, and environmental domains of QoL. Furthermore, nature proximity predicted greater environmental QoL; however, nature exposure had effect on physical and psychological domains, suggesting that having natural surroundings in the living place is important for environmental QoL, yet it is not enough to increase the physical and psychological QoL. In other words, for environmental QoL, it is important to have natural surroundings in living places; however, for psychological and physical benefits, it is important to visit and to notice nature. These findings are in accordance with previous evidence, suggesting that living in a greener neighborhood was unrelated to well-being, while visiting nature at least once a week was positively related to general health [36].

Our findings inform health promotion and public health policy, suggesting that enhancing nature exposure might help increase QoL. Strengthening connectedness to nature might be an important target of health promotion programs and effectively add to the promotion of public health and QoL. Notably, the findings of our study suggest that those targets might be equally important for rural and urban inhabitants. Increasing nature proximity in living environments might help promote public health through enhanced environmental QoL. 

One of the main limitations of the study is its cross-sectional nature, which did not let us understand the causal directions of the associations. However, the findings of the experimental studies suggest that nature exposure increases QoL, but not vice versa [74,75,76]. Next, non-probability volunteer sampling is associated with the risk of “self-selection” effect, and it is also a limitation of the present study. Another limitation of the present research is the relatively low proportion of rural inhabitants and men in the sample. However, since study measures did not differ significantly in men and women groups, and the final path model was invariant across gender groups, we consume that gender disproportion in the sample did not affect the findings. Next, the use of different-range Likert scales that were included in various instruments might also have slight effect for subjects’ answers. Further, in the present study we measured nature restorativeness as the degree of restorative outcomes after most recent visit to a natural environment. However, the measure of stress level tested general psychological stress. This might affect results of the study. It is recommended for future studies to use instruments measuring similar status of restorativeness and stress. Finally, we assessed nature exposure with items that measure subjective perception of nature contact. This did not allow us to understand what “minimal dose” of nature exposure might be needed to increase QoL. Future studies are recommended that use instruments that allow the “dose and response” associations between nature exposure and its outcomes to be better understood. 

## 5. Conclusions

Nature exposure was associated with enhanced QoL through nature restorativeness and lower stress. Nature connectedness mediated the relationships between nature exposure and greater QoL. Lower stress level was a mediator between nature exposure and QoL. Nature exposure mediated associations between nature proximity and enhanced QoL. Nature restorativeness positively predicted physical, psychological, social, and environmental domains of QoL, while connectedness to nature had positive effects on psychological and environmental domains of QoL. An independent effect of nature proximity on a better rating of the environmental QoL domain was observed. Enhancing nature exposure and nature connectedness might help strengthen QoL in urban and rural inhabitants.

## Figures and Tables

**Figure 1 ijerph-19-02098-f001:**
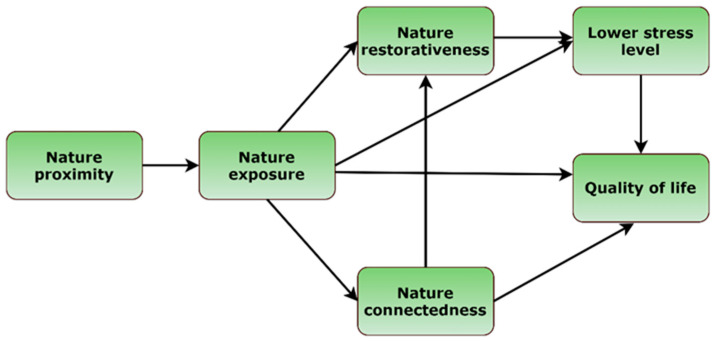
Theoretical model of the mediating effect of nature restorativeness, stress level, and nature connectedness in the association between nature exposure and quality of life.

**Figure 2 ijerph-19-02098-f002:**
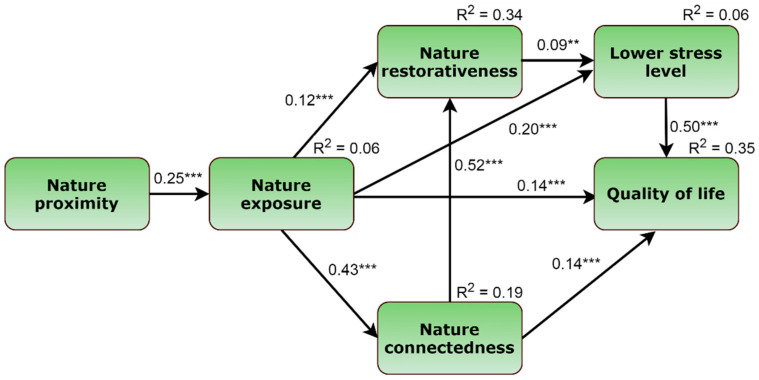
The final path model (CFI = 0.981; RMSEA = 0.061; 90% CI = 0.038–0.085). ** *p* < 0.01; *** *p* < 0.001.

**Table 1 ijerph-19-02098-t001:** Comparison of the study variables (M ± SD) across urban vs. rural place of residence groups (n = 924).

Variables	Urban n = 646	Rural n = 278	*t*-Test	Cohen’s *d*	*p*
Nature proximity	3.45 ± 1.13	3.96 ± 1.23	−6.08	−0.44	<0.001
Nature exposure	4.02 ± 0.72	4.28 ± 0.72	−5.12	−0.37	<0.001
Nature restorativeness	5.40 ± 1.47	5.40 ± 1.57	0.01	0.001	0.99
Connectedness to nature	3.80 ± 0.71	3.84 ± 0.73	−0.86	−0.06	0.39
Stress level	13.32 ± 4.29	13.63 ± 4.59	0.91	0.007	0.338
Quality of life, a total score	69.71 ± 13.11	67.66 ± 15.26	1.94	0.15	0.053
Physical domain	73.32 ± 15.02	69.82 ± 17.31	2.93	0.22	0.002
Psychological domain	67.28 ± 16.07	65.87 ± 18.18	1.12	0.08	0.265
Social domain	64.46 ± 22.71	63.52 ± 22.68	0.58	0.04	0.563
Environmental domain	70.54 ± 14.85	68.71 ± 17.06	1.55	0.12	0.121

Note: M = mean, SD = standard deviation, *p* = significance level.

**Table 2 ijerph-19-02098-t002:** Correlations between study variables (n = 924).

Variables	NP	NE	RS	CN	Lower SL	QoL
Nature proximity (NP)		0.28 ***	0.15 *	0.18 **	0.17 **	0.24 ***
Nature exposure (NE)	0.19 ***		0.34 ***	0.42 ***	0.26 ***	0.29 ***
Restorativeness (RS)	0.06	0.35 ***		0.62 ***	0.23 ***	0.34 ***
Connectedness to nature (CN)	0.07	0.44 ***	0.55 ***		0.19 **	0.28 ***
Lower stress level (SL)	0.10 *	0.24 ***	0.13 **	0.11 **		0.63 ***
Quality of life (QoL)	0.16 ***	0.35 ***	0.27 ***	0.26 ***	0.50 ***	

Note: * *p* < 0.05; ** *p* < 0.01; *** *p* < 0.001. In the upper diagonal, correlations for the rural inhabitants group (n = 278) are presented, while correlations for the urban (n = 646) inhabitants group are presented in the lower diagonal.

**Table 3 ijerph-19-02098-t003:** Multiple linear regression models to predict the domains of quality of life by the study variables (n = 924).

Variables	Physical Domain			Psychological Domain			Social Domain			Environmental Domain		
	B	β	*p*	B	β	*p*	B	β	*p*	B	β	*p*
Nature proximity	0.71	0.05	0.071	0.70	0.05	0.079	0.58	0.03	0.352	1.60	0.12	<0.001
Nature exposure	1.89	0.09	0.008	2.57	0.11	<0.001	0.85	0.03	0.453	3.04	0.14	<0.001
Nature restorativeness	1.08	0.10	0.003	1.34	0.12	<0.001	1.79	0.12	0.002	0.96	0.09	0.008
Connectedness to nature	1.03	0.05	0.188	1.85	0.08	0.021	0.57	0.02	0.645	1.77	0.08	0.024
Lower stress level	1.61	0.45	<0.001	1.76	0.46	<0.001	1.55	0.30	<0.001	1.24	0.35	<0.001
Urban place of residence	3.90	0.11	<0.001	1.98	0.05	0.051	1.00	0.02	0.525	3.13	0.09	0.002
Model summary	R = 0.54; R^2^ = 0.29			R = 0.58; R^2^ = 0.34			R = 0.36; R^2^ = 0.13			R = 0.52; R^2^ = 0.27		

Note: Models are adjusted for place of residence (urban vs. rural); B = unstandardized regression coefficient, β = standardized regression coefficient, *p* = level of significance, R² = non-adjusted R square.

**Table 4 ijerph-19-02098-t004:** Indirect effects with standard error (SE) and 95% confidence intervals in the final path model.

Pathway	Unstd.	95% CI
Nature proximity → nature exposure → lower stress level	0.21 (0.04)	0.13–0.31
Nature proximity → nature exposure → nature connectedness	0.06 (0.01)	0.05–0.09
Nature proximity → nature exposure → nature restorativeness	0.11 (0.02)	0.07–0.15
Nature proximity → nature exposure → nature connectedness and lower stress level → quality of life	0.90 (0.16)	0.62–1.26
Nature exposure → nature connectedness → nature restorativeness	0.46 (0.04)	0.38–0.55
Nature exposure → nature restorativeness → lower stress level	0.19 (0.08)	0.04–0.34
Nature exposure → nature connectedness and lower stress level → quality of life	3.34 (0.43)	2.43–4.14

Note: Unstd. = Unstandardized effect, CI = confidence interval.

## Data Availability

The dataset generated and analyzed during the current study is not publicly available but is available from the corresponding author upon reasonable request.

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
