# Peer review of "The Mediating Effect of Nature Restorativeness, Stress Level, and Nature Connectedness in the Association between Nature Exposure and Quality of Life"

_ijerph, 2022, doi:10.3390/ijerph19042098_

Round 1

Reviewer 1 Report

This study examines the overall relationship between proximity to nature, nature connectedness, etc., and stress level and QoL. Many aspects of these relationships have not been fully explored, so in my opinion, this kind of study is worthwhile. Still, several points are unclear or questionable about the survey methodology, analysis, and interpretation of the results.

First of all, the sample size bias between men and women is quite large. If there were significant differences between men and women in the relationships addressed in this study, it would create problems in generalizing the results. Also, is the ratio of men to women similar in urban and rural areas? If the ratio of males to females is significantly different in the two groups, it would further limit the interpretation of the results.

Then, I have a few questions about the scale used in this study. I recognize that the ROS is a scale to measure the sense of recovery from some specific nature experiences. However, this study does not seem to have surveyed people after having specific nature experiences. If this is the case, what do this scale's measurements in this study represent? Please clarify how you instructed the respondents in responding to the scale.

In addition, it does not seem appropriate to call ROS scores "nature restorativeness". If they were not measured immediately after some nature activities, the exact source of the restorative experience was uncertain.

Regarding the analysis, line 279 says "controlling for the residence," but how did you control it in the regression analysis? Please clarify the method. Also, I think Table "2" in line 280 is a mistake for Table "3."

There are also some problems with the path analysis. In my understanding, Ulrich's theory states that positive emotions elicited by exposure to nature will lead to stress reduction. However, the current study did not measure positive emotions when exposed to nature, and it would be difficult to verify Ulrich's theory from this data. As for the stress levels measured in this study, the RSI values are general stress values, not the values during or immediately after the nature experience. It seems unreasonable to me to examine the immediate psychological processes that occur during nature experiences from survey data that indicate a general or a long-span state.

Regarding the analysis model shown in figure 2, if the concept of QoL is so broad that it is necessary to examine each domain separately, it would be better to distinguish the domains in the model as well.

Minor issues:

Table 1: There is a description of CI in the notes, but the CI values are not in the table.

Table 2: The sample sizes for rural and urban in the notes should be reversed.

Table 4: Is this table necessary? I think it is a duplication of information since it can be calculated from the coefficients shown in Figure 2.

Author Response

Thank you for your time reviewing our paper and for your valuable comments. Please find our comments and explanations. Please see the pdf attachment.

Reviewer 2 Report

Liebe Autoren,

vielen Dank für die Möglichkeit des Reviews ihres interessanten Manuskripts.  Das Studiendesign und die theoretische Aufbereitung der Variablen ist gut gelungen. Meine beiden Hauptkritikpunkte betreffen die Validierung der Skalen und die Vorgehensweise bei der Berechnung des Pfadmodells (siehe ausführliche Kommentare in der PDF-Dateien des Manuskripts. Ich würde Sie bitte in Ruhe meine Kommentare zu lesen und diese einzuarbeiten.

Viele Grüße und viel Erfolg bei der Publikation ihrer Ergebnisse!

Author Response

(The authors gave the same response as above.)

Round 2

Reviewer 1 Report

The comments I made have been adequately addressed, and the revision clarifies the methods and results. I now recommend accepting this paper.